# Graph Autoencoders with Deconvolutional Networks

## Abstract

Recent studies have indicated that Graph Convolutional Networks (GCNs) act as a *low pass* filter in spectral domain and encode smoothed node representations. In this paper, we consider their opposite, namely Graph Deconvolutional Networks (GDNs) that reconstruct graph signals from smoothed node representations. We motivate the design of Graph Deconvolutional Networks via a combination of inverse filters in spectral domain and de-noising layers in wavelet domain, as the inverse operation results in a *high pass* filter and may amplify the noise. Based on the proposed GDN, we further propose a graph autoencoder framework that first encodes smoothed graph representations with GCN and then decodes accurate graph signals with GDN. We demonstrate the effectiveness of the proposed method on several tasks including unsupervised graph-level representation , social recommendation and graph generation.

## 1 Introduction

Autoencoders have demonstrated excellent performance on tasks such as unsupervised representation learning (Bengio, 2009) and de-noising (Vincent et al., 2010). Recently, several studies (Zeiler & Fergus, 2014; Long et al., 2015) have demonstrated that the performance of autoencoders can be further improved by encoding with Convolutional Networks and decoding with Deconvolutional Networks (Zeiler et al., 2010). Notably, Noh et al. (2015) present a novel symmetric architecture that provides a bottom-up mapping from input signals to latent hierarchical feature space with {convolution, pooling} operations and then maps the latent representation back to the input space with {deconvolution, unpooling} operations. While this architecture has been successful when processing features with structures existed in the Euclidean space (e.g., images), recently there has been a surging interest in applying such a framework on non-Euclidean data like graphs. However, extending this autoencoder framework to graph-structured data requires Graph Deconvolutional operations, which remains open-ended and hasn't been well-studied as opposed to the large body of works that have already been proposed for Graph Convolutional Networks (Defferrard et al., 2016; Kipf & Welling, 2017). In this paper, we study the characteristics of Graph Deconvolutional Networks (GDNs), and observe de-noising to be the key for effective deconvolutional operations. Therefore, we propose a wavelet-based module (Hammond et al., 2011) that serves as a de-noising mechanism after the signals reconstructed in the spectral domain (Shuman et al., 2013) for deconvolutional networks.

Most GCNs proposed by prior arts, e.g., Cheby-GCN (Defferrard et al., 2016) and GCN (Kipf & Welling, 2017), exploit spectral graph convolutions (Shuman et al., 2013) and Chebyshev polynomials (Hammond et al., 2011) to retain coarse-grained information and avoid explicit eigendecomposition of the graph Laplacian. Until recently, Wu et al. (2019) and Donnat et al. (2018) have noticed that GCN acts as a *low pass* filter in spectral domain and retains smoothed representations. Inspired by prior arts in the domain of signal deconvolution (Kundur & Hatzinakos, 1996), we propose to design a GDN by using *high pass* filters as the counterpart of *low pass* filters embodied in GCNs. Due to the nature of signal deconvolution being ill-posed, several prior arts (Donoho & Johnstone, 1994; Figueiredo & Nowak, 2003) rely on transforming these signals into another domain (e.g., spectral domain) where the problem can be better posed and resolved. Furthermore, Neelamani et al. (2004) observe inverse filters in spectral domain may amplify the noise, and we observe the same phenomenon for GDNs. Therefore, inspired by their proposed hybrid spectral-wavelet method—inverse signal reconstruction in spectral domain followed by a de-noising step in

wavelet domain—we introduce a spectral-wavelet GDN to decode the smoothed representations into the input graph signals. The proposed spectral-wavelet GDN employs spectral graph convolutions with a *high pass* filter to obtain inversed signals and then de-noises the inversed signals in wavelet domain. In addition, we apply Maclaurin series as a fast approximation technique to compute both *high pass* filters and wavelet kernels (Donnat et al., 2018).

With the proposed spectral-wavelet GDN, we further propose a graph autoencoder (GAE) framework that resembles the symmetric fashion of architectures (Noh et al., 2015). We then evaluate the effectiveness of the proposed GAE framework with three popular and important tasks: unsupervised graph-level representation (Sun et al., 2020), social recommendation (Jamali & Ester, 2010) and graph generation. In the first task, the proposed GAE outperforms the state-of-the-arts on graph classification in an unsupervised fashion, along with a significant improvement on running time. In the second task, the performance of our proposed GAE is on par with the state-of-the-arts on the recommendation accuracy; at the meantime, the proposed GAE demonstrates strong robustness against rating noises and achieves the best recommendation diversification (Ziegler et al., 2005). In the third task, our proposed GDN can enhance the generation performance of popular variational autoencoder frameworks including VGAE (Kipf & Welling, 2016) and Graphite (Grover et al., 2019).

## 2 RELATED WORK

**Deconvolutional networks**   The area of signal deconvolution (Kundur & Hatzinakos, 1996) has a long history in the signal processing community and is about the process of estimating the true signals given the degraded or smoothed signal characteristics (Banham & Katsaggelos, 1997). Later deep learning studies (Zeiler et al., 2010; Noh et al., 2015) consider deconvolutional networks as the opposite operation for Convolutional Neural Networks (CNNs) and have mainly focused on Euclidean structures, e.g., image. Some work (Dumoulin & Visin, 2016) notices Zeiler et al. (2010) is in essence a transposed convolution network as it differs from what is used in the signal processing community. For deconvolutional networks in non-Euclidean structures like graphs, the study is still sparse. Feizi et al. (2013) propose the network deconvolution as inferring the true network given partially observed structure. It relies on explicit eigen-decomposition and cannot be used as the counterpart for GCN. Yang & Segarra (2018) formulate the deconvolution as a pre-processing step on the observed signals, in order to improve classification accuracy. Zhang et al. (2020) consider recovering graph signals from the latent representation. However, it just adopts the filter design used in GCN and sheds little insight into the internal operation of GDN.

**Graph autoencoders**   Since the introduction of Graph Neural Networks (GNNs) (Kipf & Welling, 2017; Defferrard et al., 2016) and autoencoders (AEs), many studies (Kipf & Welling, 2016; Grover et al., 2019) have used GNNs and AEs to encode to and decode from latent representations. Recently *graph pooling* has emerged as a research topic that also contributes to the development of graph autoencoders. Common practices include DIFFPOOL (Ying et al., 2018), SAGPool (Lee et al., 2019), MinCut-Pool (Bianchi et al., 2020). Although some encouraging progress has been achieved, there is still no work about graph deconvolution that can up-sample latent feature maps to restore their original resolutions (Gao & Ji, 2019). In this regard, current graph autoencoders bypass the difficulty via (1) non-parameterized decoders (Kipf & Welling, 2016; Deng et al., 2020; Li et al., 2020), (2) GCN decoders (Grover et al., 2019; Gao & Ji, 2019), and (3) multilayer perceptron (MLP) decoders (Simonovsky & Komodakis, 2018).

## 3 GRAPH AUTOENCODER FRAMEWORK

Formally, we are given an undirected, unweighted graph $G = (V, A, X)$. $V$ is the node set and $N = |V|$ denotes the number of nodes. The adjacency matrix $A \in \mathbb{R}^{N \times N}$ represents the graph structure. The feature matrix $X \in \mathbb{R}^{N \times d}$ represents the node attributes. Our goal is to learn an encoder and a decoder to map between the space of graph $G$ and their latent factors $G^{pool} = (V^{pool}, A^{pool}, Z)$. We show a schematic diagram of our proposed framework in Figure 1.

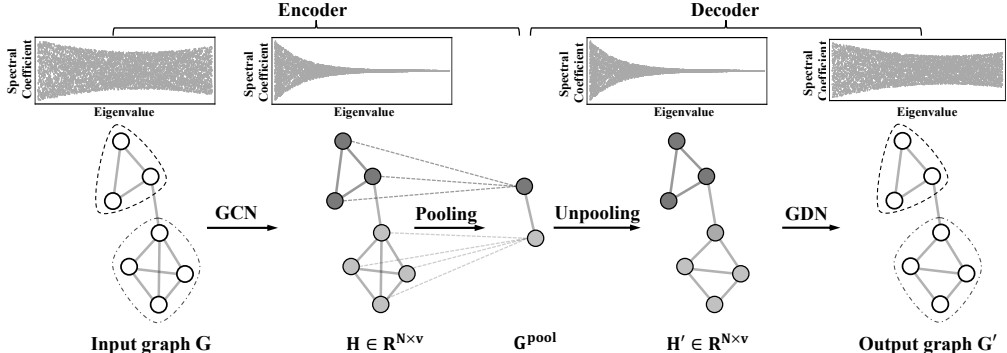

Figure 1: Schematic diagram of the proposed graph autoencoder framework. We plot graphs in spatial domain together with their signal coefficients in spectral domain. In the encoding stage, we run a GCN model to derive smoothed node representations and obtain coarse-grained graph via a pool layer. In the decoding stage, we upscale the coarsened graph back and reconstruct the graph signal via a GDN model.

### 3.1 ENCODER

Our encoder consists of several layers of Graph Convolutional Networks (GCNs) (Kipf & Welling, 2017) and a pooling layer, to produce coarser representations of the input graphs.

**Convolution**  The convolutional layers are used to derive smoothed node representations, such that nodes that are similar in topological space should be close enough in Euclidean space.

$$H = \text{GCN}(A, X), \tag{1}$$

where $H \in \mathbb{R}^{N \times v}$ denotes smoothed node representations. Specifically, Wu et al. (2019) show that GCN is a *low pass* filter in spectral domain with $g_c(\lambda_i) = 1 - \lambda_i$, where $\{\lambda_i\}_{i=1}^N$ are the eigenvalues of the normalized Laplacian matrix $L_{sym} = D^{-\frac{1}{2}} L D^{-\frac{1}{2}}$, $L$ and $D$ are the Laplacian and degree matrices of the input graph $A$ respectively.

**Pooling**  We follow Lee et al. (2019) and Li et al. (2019) by using the self attention mechanism to pool the fine-grained graph into coarse-grained representations,

$$S = \text{softmax}\big(\tanh(HW_1)W_2\big), \tag{2}$$

where $W_1 \in \mathbb{R}^{v \times d}$ and $W_2 \in \mathbb{R}^{d \times K}$ are two weight matrices, $W_1$ is used for feature transformation and $W_2$ is used to infer the membership of each node with respect to each cluster $V_k$.

Similar to Ying et al. (2018), we compute the coarsed graph structure $A^{pool} \in \mathbb{R}^{K \times K}$ and feature representation $Z \in \mathbb{R}^{K \times v}$ as follows:

$$Z = S^\top H; \qquad A^{pool} = S^\top A S. \tag{3}$$

Note here $(Z, A^{pool})$ is size invariant and permutation invariant, as pointed out by Li et al. (2019).

### 3.2 DECODER

Our decoder consists of an unpooling layer and several layers of Graph Deconvolutional Networks (GDNs), to produce fine-grained graphs from the encoded $G^{pool}$.

**Unpooling**  We follow Bianchi et al. (2020) to upscale the coarsened graph back to its original size,

$$H' = SZ; \qquad A' = SA^{pool}S^\top. \tag{4}$$

**Deconvolution**  The deconvolutional layers are used to recover the original graph features given smoothed node representations,

$$X' = \text{GDN}(A', H'), \tag{5}$$

where $X' \in \mathbb{R}^{N \times d}$ denotes the recovered graph features. We shall further discuss our design of GDN in Section 4.

### 3.3 THE LOSS FUNCTION

The overall reconstruction loss is a weighted sum of structure reconstruction loss and feature reconstruction loss.

$$\mathcal{L} = \lambda_A f(A, A') + \lambda_X f(X, X'), \tag{6}$$

where $f(\cdot, \cdot)$ denotes a differential distance metric, e.g., $f(\cdot, \cdot) = \text{MSE}(\cdot, \cdot)$ for continuous input, and $\text{MSE}(\cdot, \cdot)$ represents mean squared error.

## 4 GRAPH DECONVOLUTIONAL NETWORKS

In this section, we present our design of Graph Deconvolutional Networks (GDNs). A naive deconvolutional nets can be obtained using the inverse operator $g_c^{-1}(\lambda_i) = \frac{1}{1-\lambda_i}$ in spectral domain. Unfortunately, inverse operation results in a *high pass* filter and may amplify the noise (Donoho & Johnstone, 1994; Figueiredo & Nowak, 2003). In this regard, we propose an efficient, hybrid spectral-wavelet deconvolutional network that performs inverse signal recovery in spectral domain first, and then conducts a de-noising step in wavelet domain to remove the amplified noise (Neelamani et al., 2004).

### 4.1 INVERSE OF GCN

In order to recover graph signals from the latent representation computed by GCN encoder, we proposed a naive approach–inverse GCN with the inverse filter as $g_c^{-1}(\lambda_i) = \frac{1}{1-\lambda_i}$ in spectral domain. The spectral graph convolution on a signal $x \in \mathbb{R}^N$ is defined as:

$$g_c^{-1} * x = U \text{diag}(g_c^{-1}(\lambda_1), \ldots, g_c^{-1}(\lambda_N)) U^\top x = U g_c^{-1}(\Lambda) U^\top x, \tag{7}$$

where $U$ is the eigenvector matrix of the normalized graph Laplacian $L_{sym} = U \Lambda U^\top$. Then, we apply Maclaurin series approximation on $g_c^{-1}(\Lambda) = \sum_{n=0}^{\infty} \Lambda^n$ and get a fast algorithm as below:

$$g_c^{-1} * x = U \sum_{n=0}^{\infty} \Lambda^n U^\top x = \sum_{n=0}^{\infty} L_{sym}^n x. \tag{8}$$

As in GCN (Kipf & Welling, 2017), when the first order approximation is used to address overfitting, we derive a spectral filter with $g_c^{-1}(\lambda_i) = 1 + \lambda_i$, which is apparently a *high pass* filter.

Following GCN (Kipf & Welling, 2017), a feature transformation is applied to increase filter strength. Recap the GDN in Section 3.2, the inverse version of GCN can be written as:

$$M = (I_N + L'_{sym}) H' W_3, \tag{9}$$

where $L'_{sym}$ is the corresponding normalized graph Laplacian matrix for $A'$, $H'$ is the smoothed representations and $W_3$ is the parameter set to be learned.

Compared with directly using GCN for signal reconstruction in Zhang et al. (2020), the proposed inverse GCN demonstrates its efficacy in recovering the high frequency signals of the graph, as shown in Figure 2 (b) and (d). We shall further discuss this point in Section 4.3.

### 4.2 WAVELET DE-NOISING

The proposed inverse GCN may over-amplifies the high frequency signals and introduce undesirable noise into the output graph. Thus, a de-noising process is necessary to separate the useful information and the noise. Wavelet-based methods have a strong impact on the field of de-noising, especially in image restoration when noise is amplified by inverse filters (Neelamani et al., 2004). Notably, coefficients in wavelet domain could allow the noise to be easily separated from the useful information, while transformation into spaces such as spectral domain does not share the same characteristics. In the literature, many wavelet de-noising methods have been proposed, e.g., SureShrink (Donoho & Johnstone, 1994), BayesShrink (Chang et al., 2000), and they differ in how they estimate the separating threshold. Our method generalizes these threshold ideas and automatically separates the noise from the useful information with a learning-based approach.

Consider a set of wavelet bases $\Psi_s = (\Psi_{s1}, \ldots, \Psi_{sN})$, where each one $\Psi_{si}$ denotes a signal on graph diffused away from node $i$ and $s$ is a scaling parameter (Xu et al., 2019a), the wavelet bases

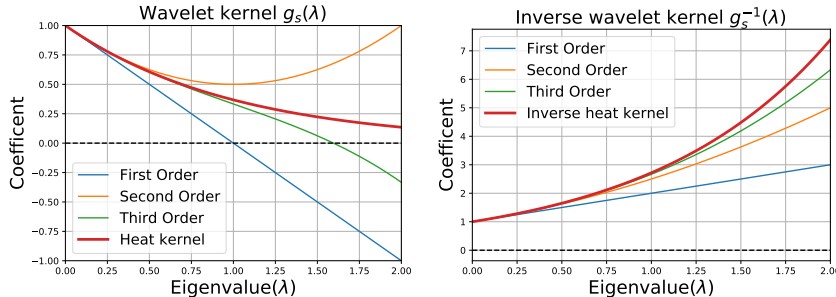

Figure 2: Low-order Maclaurin Series well approximate wavelet kernel and inverse wavelet kernel with $s = 1$.

can be written as $\Psi_s = U g_s(\Lambda) U^\top$ and $g_s(\cdot)$ is a filter kernel. Following previous works GWNN (Xu et al., 2019a) and GRAPHWAVE (Donnat et al., 2018), we use the heat kernel $g_s(\lambda_i) = e^{-s\lambda_i}$, as heat kernel admits easy calculation of inverse wavelet transform with $g_s^{-1}(\lambda_i) = e^{s\lambda_i}$. In addition, we can apply Maclaurin series approximation on heat kernel and neglect explicit eigendecomposition by:

$$\Psi_s = U \sum_n \frac{(-1)^n}{n!} s^n \Lambda^n U^\top = \sum_{n=0}^{\infty} \frac{(-1)^n}{n!} s^n L_{sym}^n, \tag{10}$$

$$\Psi_s^{-1} = U \sum_n \frac{s^n}{n!} \Lambda^n U^\top = \sum_{n=0}^{\infty} \frac{s^n}{n!} L_{sym}^n, \tag{11}$$

Usually truncated low order approximation is used in practice with $n \leq 3$ (Defferrard et al., 2016).

To cope with the noise threshold estimation problem, we introduce a parameterized matrix $W_2$ in wavelet domain and eliminate these noise coefficients via a ReLU function, in the hope of learning a separating threshold from the data itself. Take the noised representation $M$ raised in Section 4.1 into consideration, we derive the reconstructed signal as:

$$X' = \Psi_s \text{ReLU}(\Psi_s^{-1} M W_4) W_5. \tag{12}$$

where $W_4$ and $W_5$ are trainable parameters.

We then contrast the difference between Wavelet Neural Network (WNN) in this work and other related WNNs including GWNN (Xu et al., 2019a) and GRAPHWAVE (Donnat et al., 2018).

**Scalability**  An important issue in WNN is that one needs to avoid explicit eigendecomposition to compute wavelet bases for large graphs. Both GRAPHWAVE and GWNN, though try to avoid eigendecomposition by exploiting Chebyshev polynomials, still rely integral operations (see Appendix D in Xu et al. (2019a)) and there is no clear way that we can scale up. Differently, we use Maclaurin series to approximate wavelet bases, which has explicit polynomial coefficients and can resemble the heat kernel well when $n = 3$. Please refer to Figure 2 for more details.

**De-noising**  The purpose of both GWNN and GRAPHWAVE is to derive node presentations with localized graph convolution and flexible neighborhood, such that downstream tasks like classification can be simplified. On the contrary, our work implements wavelet neural network in the purpose of detaching the useful information and the noise amplified by inverse GCN. Due to the different purpose, our work applies the activation function in wavelet domain while GWNN in the original vertex domain.

### 4.3 VISUALIZATION

In Figure 3, we illustrate the difference between GDN and each component inside by visualizing the reconstructed road occupancy rates in a traffic network. The traffic network targets District 7 of California collected from Caltrans Performance Measurement System (PeMS)[1]. We select 4438

---

[1]http://pems.dot.ca.gov/

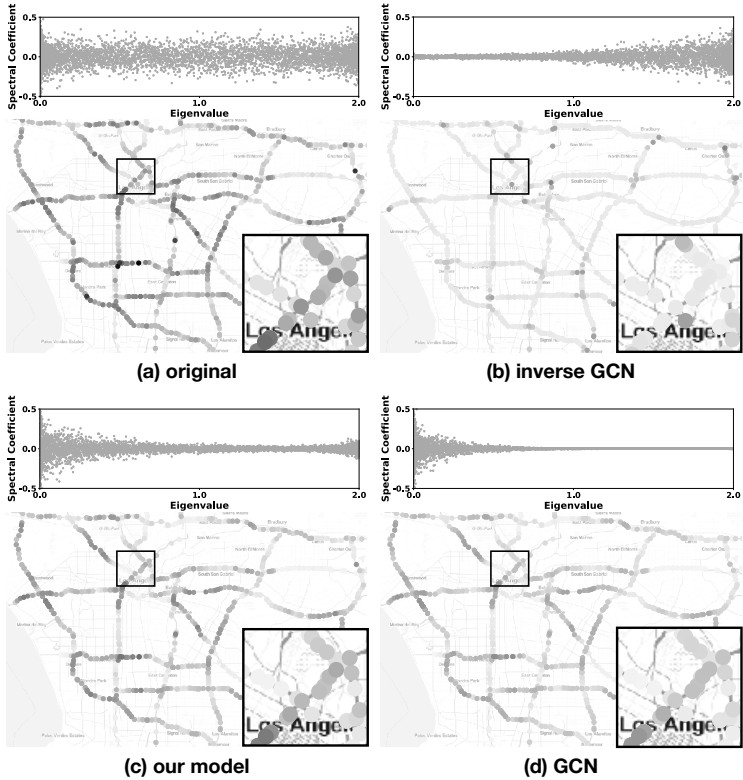

Figure 3: Illustration of road occupancy rates decoded by different approaches at 5pm on June 24, 2020 in District 7 of California. A darker color implies a higher road occupancy rate. In each figure above, we plot the reconstructed signal in spatial domain together with coefficients in spectral domain. (a) the original, (b) inverse GCN (RMSE = 0.11), (c) GDN/our model (RMSE = 0.07), (d) GCN (RMSE = 0.09).

sensor stations as the node set $V$ and collect their road average occupancy rates at 5pm on June 24, 2020. Following Li et al. (2018), we construct an adjacency matrix A by denoting $A_{ij} = 1$ if sensor station $v_j$ and $v_i$ are adjacent on a freeway along the same direction. We reconstruct road occupancy rate of each sensor station with three decoder variants: (b) inverse GCN, (c) GDN, (d) GCN. Here the three variants share the same encoders defined in Section 3.1.

As can be seen, while both GDN and GCN decoders can resemble the input signal in low frequency, GDN decoder retains more high frequency information. For inverse GCN decoder, although keeping much high frequency information (mixed with noise), it drops lots of low frequencies, making the decoded signals less similar to the input in Euclidean space.

## 5 EXPERIMENTS

We first validate both the proposed graph autoencoder framework and GDN in two tasks including unsupervised graph-level representation (Sun et al., 2020; Hassani & Khasahmadi, 2020) and social recommendation (Jamali & Ester, 2010; Monti et al., 2017; Berg et al., 2018), we then test GDN in graph generation tasks (Kipf & Welling, 2016; Grover et al., 2019).

### 5.1 UNSUPERVISED GRAPH-LEVEL REPRESENTATION

Following recent methods (Sun et al., 2020; Hassani & Khasahmadi, 2020), we evaluate the effectiveness of the unsupervised graph-level representations on downstream graph classification tasks.

Table 1: Mean 10-fold cross validation accuracy on five graph datasets for kernel and unsupervised methods

| | Datasets | IMDB-BIN | IMDB-MULTI | REDDIT-BIN | PROTEINS | DD |
|---|---|---|---|---|---|---|
| Kernel | RW (Gärtner et al., 2003) | 50.7 ±0.3 | 34.7 ±0.2 | - | 74.2 ±0.4 | - |
| | SP (Borgwardt & Kriegel, 2005) | 55.6 ±0.2 | 38.0 ±0.3 | 64.1 ±0.1 | 75.07 ±0.5 | 78.7 ±3.9 |
| | GK (Shervashidze et al., 2009) | 65.9 ±1.0 | 43.9 ±0.4 | 77.3 ±0.2 | 71.67 ±0.6 | 74.9 ±3.8 |
| | WL (Shervashidze et al., 2011) | 72.3 ±3.4 | 47.0 ±0.5 | 68.8 ±0.4 | 72.92 ±0.6 | 76.4 ±2.4 |
| | DGK (Yanardag & Vishwanathan, 2015) | 67.0 ±0.6 | 44.6 ±0.5 | 78.0 ±0.4 | 75.7 ±0.5 | - |
| | MLG (Kondor & Pan, 2016) | 66.6 ±0.3 | 41.2 ±0.0 | - | **76.3 ±0.7** | - |
| Supervised | GCN (Kipf & Welling, 2017) | 74.0 ±3.4 | 51.9 ±3.8 | 50.0 ±0.0 | 76.0 ±3.2 | - |
| | GAT (Veličković et al., 2018) | 70.5 ±2.3 | 47.8 ±3.1 | 85.2 ±3.3 | - | - |
| | GIN-0 (Xu et al., 2019b) | 75.1 ±5.1 | **52.3 ±2.8** | **92.4 ±2.5** | 76.2 ±2.8 | - |
| | GIN-$\epsilon$ (Xu et al., 2019b) | 74.3 ±5.1 | 52.1 ±3.6 | 92.2 ±2.3 | 75.9 ±3.8 | - |
| Unsupervised | VGAE (Kipf & Welling, 2016) | 64.9±0.38 | 38.9±0.46 | - | 72.4±0.42 | 76.3±0.34 |
| | SUB2VEC (Adhikari et al., 2018) | 55.3 ±1.5 | 36.7 ±0.8 | 71.5 ±0.4 | 73.3 ±2.1 | - |
| | GRAPH2VEC (Narayanan et al., 2017) | 71.1 ±0.5 | 50.4 ±0.9 | 75.8 ±1.0 | 73.3 ±2.1 | - |
| | INFOGRAPH (Sun et al., 2020) | 73.0 ±0.9 | 49.7 ±0.5 | 82.5 ±1.4 | 75.0 ±1.3 | 76.9 ±1.3 |
| | MVGRL (Hassani & Khasahmadi, 2020) | 74.2 ±0.7 | 51.2 ±0.5 | 84.5 ±0.6 | 75.9 ±1.9 | 78.3 ±1.7 |
| | ALATION-INVERSE-GCN | 73.2 ±1.9 | 50.4 ±1.3 | 83.9 ±1.3 | 75.0 ±1.2 | 77.8 ±1.3 |
| | ALATION-GCN | 74.2 ±1.4 | 49.6 ±1.2 | 84.6 ±1.2 | 74.9 ±1.2 | 77.9 ±1.0 |
| | OURS | **76.0 ±1.3** | 51.5 ±1.4 | 86.0 ±1.7 | 76.1 ±1.4 | **79.1 ±1.5** |

**Data and baselines** We use five graph classification benchmarks including IMDB-Binary, IMDB-Multi, Reddit-Binary, PROTEINS and DD (Narayanan et al., 2017; Ying et al., 2018). For the detailed statistics, please refer to Table 4. We compare with six graph kernels: Random Walk (RW) (Gärtner et al., 2003), Shortest Path Kernel (SP) (Borgwardt & Kriegel, 2005), Graphlet Kernel (GK) (Shervashidze et al., 2009), Weisfeiler-Lehman Sub-tree Kernel (WL) (Shervashidze et al., 2011), Deep Graph Kernels (DGK) (Yanardag & Vishwanathan, 2015) and Multi-Scale Laplacian Kernel (MLG) (Kondor & Pan, 2016). In addition, we compare with four unsupervised graph-level representation learning methods: SUB2VEC (Adhikari et al., 2018), GRAPH2VEC (Narayanan et al., 2017), INFOGRAPH (Sun et al., 2020) and MVGRL (Hassani & Khasahmadi, 2020). We also include the results of recent supervised graph classification models: GCN (Kipf & Welling, 2017), GAT (Veličković et al., 2018), GIN (Xu et al., 2019b). We denote our framework using (1) GCN (Kipf & Welling, 2017) in the decoders as ALATION-GCN[2], (2) inverse of GCN in Section 4.1 in the decoders as ALATION-INVERSE-GCN.

**Setup** We adopt the same procedure of previous works (Sun et al., 2020; Hassani & Khasahmadi, 2020) and report the mean 10-fold cross validation accuracy with standard deviation after 5 runs using LIBSVM (Chang & Lin, 2011). We report results from previous papers with the same setup if available. For the datasets PROTEINS and DD, we implement strong baselines including INFO-GRAPH and MVGRL, with a hyperparameter search according the papers.

We train our model using minibatch based Adam optimizer with a learning rate of 0.01. We use the cross-entropy loss to reconstruct the features. In our encoders, the best variants of GNN are chosen from GCN (Kipf & Welling, 2017) and Heatts (Li et al., 2020). For Heatts, we let $s = 1$ for all experiments. For detailed hyperparameter settings, please refer to Appendix C.

**Results** The classification accuracies on the five benchmarks are shown in Table 4. MLG, as a kernel method, performs well on PROTEINS. However, it suffers from a long run time and takes more than 1 day on two larger datasets, as observed in INFOGRAPH. Our method achieves the best results in 4 out of 5 datasets compared with both kernel and unsupervised models, e.g., it achieves 1.5% improvement over previous state-of-the-art on Reddit-Binary, and 1.8% improvement on IMDB-Binary, which shows the superiority of our methods. When compared with supervised graph classification models, ours beats the best supervised classification model GIN on IMDB-BIN, on-par with GIN on PROTEINS, IMDB-MULTI, and only loses on REDDIT.

---

[2]note ALATION-GCN coincides with Zhang et al. (2020).

Table 2: Comparison of different methods on social recommendation tasks

| Datasets | Ciao | | Douban | |
|---|---|---|---|---|
| - | ILS | RMSE | ILS | RMSE |
| sRGCNN (Monti et al., 2017) | 1.63% | 1.183 | 6.03% | 0.801 |
| GC-MC (Berg et al., 2018) | 1.09% | **1.061** | 3.90% | **0.734** |
| GraphRec (Fan et al., 2019) | 1.24% | 1.062 | 8.27% | 0.754 |
| OURS | **0.48%** | 1.071 | **2.65%** | 0.745 |

**Ablation study**   We investigate the role of each component in our GDN. We let the three variants share the same depth of layers and parameter size. As observed in Table 4, when decoding by pure inverse of GCN, the performance is just comparable to that of GCN decoders and outperformed by our GDN decoders in all 5 datasets, indicating the effectiveness of this hybrid design.

**Running time**   We observe that our model runs significantly faster than INFOGRAPH and MV-GRL. Our model takes 10s to train one epoch of PORTEINS on Tesla P40 24G, while INFOGRAPH needs 127s and MVGRL needs 193s. This is because our model neglects the tedious process of negative sampling used in both INFOGRAPH and MVGRL.

## 5.2   SOCIAL RECOMMENDATION

In our model, we consider social recommendation as feature recovery on graphs, i.e., user-item ratings are represented as the feature matrix and a social network among users is represented as the graph structure. The model is trained on an incomplete version of feature matrix (training data) and is used to infer the potential ratings (test data).

**Data and baselines**   We use two social recommendation benchmark datasets: Douban and Ciao[3]. Both datasets consist of user ratings for items and incorporate a social network among users. For Douban, we use the preprocessed dataset provided by Monti et al. (2017). For Ciao, we use a sub-matrix of 7,317 users and 1,000 items. Dataset statistics are summarized in Table 5 in the Appendix. We compare with three baselines: sRGCNN (Monti et al., 2017), GC-MC (Berg et al., 2018), and GraphRec (Fan et al., 2019). Following Candes & Plan (2010), we consider the few available entries (training data) should be corrupted with noise in reality, and add random ratings to the training data at different level $p \in \{0, 0.1, \ldots, 1\}$, where $p = 0$ denotes no noise is added.

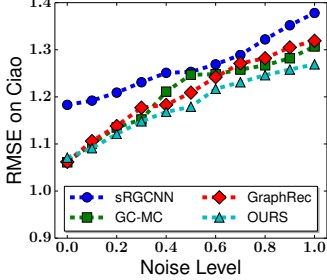
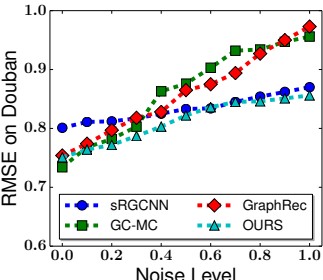

Figure 4: Comparison of different methods on social recommendation tasks. The x-axis denotes the noise level while the y-axis denotes RMSE on {Ciao, Douban}.

**Setup**   For social recommendation, we neglect pooling and unpooling operations as the predictive task is within a *single* graph. We use the Mean Squared Error (MSE) loss to reconstruct the ratings. Same as in GC-MC (Berg et al., 2018), we stack the output of the first GCN layer and the second GCN layer in our encoders, and use left normalization to preprocess adjacency matrix. For

---

[3]https://www.ciao.co.uk/

Table 3: The effect of GDN with various graph generation methods

| Datasets | **MUTAG** | | | **PTC-MR** | | | **ZINC** | | |
|---|---|---|---|---|---|---|---|---|---|
| - | $\log p(A\|Z)$ | AUC | AP | $\log p(A\|Z)$ | AUC | AP | $\log p(A\|Z)$ | AUC | AP |
| VGAE (Kipf & Welling, 2016) | -1.156 | 0.869 | 0.645 | -1.366 | 0.566 | 0.433 | -1.033 | 0.551 | 0.288 |
| VGAE with GDN | -1.114 | 0.880 | 0.678 | -1.351 | 0.760 | 0.602 | -0.997 | **0.862** | **0.613** |
| Graphite (Grover et al., 2019) | -1.140 | 0.868 | 0.632 | -1.362 | 0.564 | 0.437 | -1.043 | 0.559 | 0.288 |
| Graphite with GDN | **-1.104** | **0.882** | **0.681** | **-1.347** | **0.773** | **0.613** | **-0.989** | 0.851 | 0.593 |

recommendation accuracy, we report Root Mean Squared Error (RMSE). For recommendation diversification (Ziegler et al., 2005), we report Intra-List Similarity (ILS). For the definition of ILS and detailed hyperparameter settings, please refer to Appendix C.

**Results** The recommendation RMSE and ILS on the two benchmarks ($p = 0$) are shown in Table 2. Our method performs on a par with state-of-the-art methods on RMSE, with only lose to the best model GC-MC by 0.01 on Ciao and 0.011 on Douban. As for ILS, our method performs the best, e.g., it beats the second best model GC-MC by 0.61% on Ciao and 1.25% on douban, which shows the superiority of our methods on recommendation diversification. The recommendation RMSE with respect to different noise level is shown in Figure 4. As can be seen, our method performs the best with respect to different noise level ($p > 0$), e.g., it achieves 0.04 improvement over the second best model GC-MC on Ciao when $p = 1$, and 0.014 improvement over the second best model sRGCNN on Douban, which shows the superiority when user-item interactions are noisy.

### 5.3 GRAPH GENERATION

**Data and baselines** We use three molecular graphs: MUTAG (Kriege & Mutzel, 2012) containing mutagenic compounds, PTC-MR (Kriege & Mutzel, 2012) containing compounds tested for carcinogenicity and ZINC (Irwin et al., 2012) containing druglike organic molecules, to evaluate the performance of GDN on graph generation. As our proposed autoencoder framework is not suitable for graph generation task, we test GDN on two popular variational autoencoder framework including VGAE (Kipf & Welling, 2016) and Graphite (Grover et al., 2019). Our purpose is to validate if GDN helps with the generation performance.

**Setup** For VGAE and Graphite, we reconstruct the graph structures using their default methods and the features using GDN. The two reconstructions share the same encoders and sample from the same latent distributions. We train for 200 iterations with a learning rate of 0.01. The output dimension of the first hidden layer is 32 and that of the second-layer is 16. For MUTAG and PTC-MR, we use all the graph samples. For ZINC, we target test datasets (See Table 6 in the Appendix for the details). We use 50% samples as train set and the remaining 50% as test set.

**Results** Evaluating the sample quality of generative models is challenging (Theis et al., 2016). In this work, we validate the generative performance with the log-likelihood ($\log p(A|Z)$), area under the receiver operating characteristics curve (AUC) and average precision (AP). As shown in Table 3, GDN can improve the generative performance of both VGAE and Graphite in general, e.g., it improve the AP score of Graphite by 4.9% on MUTAG , the AUC score of VGAE by 19.4% on PTC-MR and $\log p(A|Z)$ of VGAE by 5.4% on ZINC, which shows the superiority of GDN.

## 6 CONCLUSION

In this paper, we present a symmetric graph autoencoder framework in an unsupervised way. The proposed framework relies on Graph Deconvolutional Networks (GDNs), the opposite of GCNs that recover graph signals from smoothed representations. The introduced GDN uses spectral graph convolutions with a *high pass* filter to obtain inversed signals and then de-noises the inversed signals in wavelet domain. The effectiveness of the proposed method is validated on unsupervised graph-level representation, social recommendation and graph generation tasks.

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

Table 4: Statistics of the datasets used in graph classification

| Datasets | IMDB-BIN | IMDB-MULTI | REDDIT-BIN | PROTEINS | DD |
|---|---|---|---|---|---|
| (No.Graphs) | 1000 | 1500 | 2000 | 1113 | 1178 |
| (No.Classes) | 2 | 3 | 2 | 2 | 2 |
| (Avg.Nodes) | 19.8 | 13.0 | 508.5 | 39.1 | 284.3 |
| (Avg.Edges) | 193.1 | 65.9 | 497.8 | 72.8 | 715.7 |

Table 5: Statistics of the datasets used in social recommendation

| Dataset | Users | Items | Train/Test Ratings | Rating Density | Social Connections | Social Density |
|---|---|---|---|---|---|---|
| Douban | 3,000 | 3,000 | 123,202/13,689 | 1.52% | 2,690 | 0.03% |
| Ciao | 7,317 | 1,000 | 39,279/16,892 | 0.77% | 111,781 | 0.21% |

## A  DETAILED MODEL CONFIGURATION

For the unsupervised graph-level representation experiments, we use the original graph structure in the decoders of the proposed graph autoencoder framework. The C parameter of LIBSVM (Chang & Lin, 2011) is selected from $\{10^{-3}, 10^{-2}, \ldots, 10^2, 10^3\}$. The depth of GNN layers in the encoders is set to 2. The output dimension of the first layer is chosen from $\{64, 128, 256\}$. The output dimension of the second layer is chosen from $\{16, 32\}$. The number of clusters $K$ is chosen from $\{16, 32\}$. The number of epochs is chosen from $[15, 20]$. The batch size is set to 1. We set $\lambda_A = 0$ and $\lambda_X = 1$. We don't use Dropout as it does not improve the performance. For all methods, the embedding dimension is set to 512 and parameters of downstream classifiers are independently tuned using cross validation on training folds of data, in order to have a fair comparison with previous works. The best average classification accuracy is reported for all methods.

For social recommendation, we train our model using Adam optimizer with a learning rate of $\{0.005, 0.002\}$. We use the original graph structure in the decoders of the proposed graph autoencoder framework. The output dimension of the first layer is 256. The output dimension of the second layer is 128. The number of epochs is 200. We use full-batch size. We don't use Dropout as it does not improve the performance. Intra-List Similarity (ILS) of one user is defined as:

$$\text{ILS}_u = \frac{1}{2} \sum_{i_m \in L} \sum_{i_n \in L} \text{sim}(i_m, i_n), \tag{13}$$

where $L$ is the item list for the user and it is equal with the top-10 items in this work. $\text{sim}(\cdot, \cdot)$ is the similarity function and is set to cosine similarity. For the representation of items, we use the input user rating vectors.

## B  MORE RELATED WORKS

**Unsupervised graph-level representations**  Unsupervised graph-level representation techniques can be generally classified into three groups. The first group is graph kernels. Representative works include Random Walk (Gärtner et al., 2003), Shortest Path Kernel (Borgwardt & Kriegel, 2005), Graphlet Kernel (Shervashidze et al., 2009), Weisfeiler-Lehman Sub-tree Kernel (Shervashidze et al., 2011), Deep Graph Kernels (Yanardag & Vishwanathan, 2015) and Multi-Scale Laplacian Kernel (Kondor & Pan, 2016). This group of techniques works on finding informative sub-structures and computes similarities based on these sub-structures. The second group is contrastive methods. Representative works include SUB2VEC (Adhikari et al., 2018), GRAPH2VEC (Narayanan et al., 2017), INFOGRAPH (Sun et al., 2020) and MVGRL (Hassani & Khasahmadi, 2020). This group employs a scoring function, e.g., mutual information (Veličković et al., 2019), to train a model that can increase the score on positive examples and decrease the score on negative samples. It thus depends on the sophisticated positive/negative sampling strategy to boost the performance. The third group is graph autoencoders (Kipf & Welling, 2016; Grover et al., 2019). Due to the lack of powerful decoders, the performance of this kind is still underestimated.

Table 6: Statistics of the datasets used in graph generation

| Datasets | MUTAG | PTC-MR | ZINC |
|---|---|---|---|
| (No.Graphs) | 188 | 344 | 5000 |
| (Avg.Nodes) | 17.9 | 14.3 | 23.1 |
| (Avg.Edges) | 19.8 | 14.7 | 49.7 |

**Social recommendation** Social recommendations assume a social network among users and these user-user interactions are helpful in boosting the recommendation performance (Jamali & Ester, 2010). As GNNs have been proven to be capable of learning on graph data, recently recommendation methods based on GNNs have shown impressive results on recommendation accuracy. Representative works include sRGCNN (Monti et al., 2017), GC-MC (Berg et al., 2018), and GraphRec (Fan et al., 2019). Specifically, GC-MC (Berg et al., 2018) models recommendations as link predictions using graph autoencoder. Though successful, GC-MC uses an MLP decoder and fails to deal with noisy ratings (Candes & Plan, 2010).

## C  DERIVATIVE OF INVERSE GCN

$$g_c^{-1}(\lambda_i) = \frac{1}{1 - \lambda_i} = \sum_{n=0}^{\infty} \frac{\left(\frac{1}{1-\lambda_i}\right)_{\lambda_i=0}^{(n)}}{n!} \lambda_i^n \tag{14}$$

$$= \sum_{n=0}^{\infty} \frac{(-1)^n n! (-1)^n}{n!} \lambda_i^n = \sum_{n=0}^{\infty} \lambda_i^n$$

where $(n)$ denotes the $n$-th order derivative. Thus, Eq.(8) can be obtained as:

$$g_c^{-1} * x = U \text{diag}(\sum_{n=0}^{\infty} \lambda_1^n, \ldots, \sum_{n=0}^{\infty} \lambda_N^n) U^\top x = U \sum_{n=0}^{\infty} \Lambda^n U^\top x \tag{15}$$

