# OpenReview forum: "Graph Autoencoders with Deconvolutional Networks"
_ICLR.cc/2021/Conference — Reject_

### Official Review · AnonReviewer2 · 2020-10-25
**The proposed GDN to learn high quality unsupervised graph embedding.**

**Rating:** 6
**Confidence:** 4

**Review:**

Summary:

The authors proposed graph deconvolution layers (GDNs) and employ GDNs to learn graph embedding in a encoder-decoder framework. The authors performs inverse signal recovery in spectral domain and then conducts a de-noising step in wavelet domain to remove the amplified noise. The proposed method can outperform baseline methods on graph classification and social recommendation. However, there are some issuses should be solved.

Pro:
The proposed method is very novel. Encoder-decoder network is firsted employed on graph data for unsupervised learning. The writing and organization of this paper is good. The authors provide a clear and detailed describtion for the proposed method. For the decoding part, the authors proposed a de-noising step in wavelet domain to remove the amplified noise.

Cons:
In this paper, the authors employ encoder-decoder framework to learn graph embedding. Why encoder-decoder framework is needed for graph data? One option is that we can use Variational Graph Auto-Encoders[1] to learn node embedding and use SUM or AVG function to obtain graph embedding. This should be added as a baseline to show that encoder-decoder framework is better. In addition, the authors should also mention how many pooling layers are used in encoder-decoder framework. And the experiments with different number of pooling layers can help understand the proposed method.

Overall: I vote to accept this paper considering the importance of the task and the novely method proposed.

[1].Kipf, Thomas N., and Max Welling. "Variational graph auto-encoders." arXiv preprint arXiv:1611.07308 (2016).

---

> ### Author Response · Authors · 2020-11-19
> **Response to Reviewer2**
>
> We sincerely thank you for the valuable comments. Our responses to the comments are listed below.
>
> Q1. Variational Graph Auto-Encoders as baselines.
>
> A1. We appreciate the reviewer's suggestions and have added VGAE (kipf \& Welling, 2016) as baselines. We report the experimental results of VGAE with SUM functions (VGAE\&SUM). VGAE\&SUM achieves 64.9\%, 38.9\%, 72.4\% and 76.3\% on IMDB-BIN, IMDB-MULTI, PROTEINS and DD respectively; The results show our method outperforms VGAE\&SUM on all datasets. In addition, we would like to highlight that our method can reconstruct both graph structures and graph features, while VGAE\&SUM focuses on reconstructing structures.
>
> Q2. In addition, the authors should also mention how many pooling layers are used in encoder-decoder framework. And the experiments with different number of pooling layers can help understand the proposed method.
>
> A2.  We use one pooling layer and one unpooling layer in the framework. We are aware that in autoencoder framework that applies to images, a common practice is stacking multiple layers of convolution and pooling, like “CNN-pooling-CNN-pooling...”. However, current GCN still suffers from the problem of over-smoothing, which means increasing the number of pooling layers and GCN layers would deteriorate the overall performance (see [1]). In this sense, we just use one pooling layer.
>
> [1] Qimai Li, Zhichao Han, Xiao-Ming Wu. Deeper Insights into Graph Convolutional Networks for Semi-Supervised Learning, in AAAI 2018, 3538--3545.

---

### Official Review · AnonReviewer4 · 2020-10-28
**The presentation of the paper is clear for the method part. Need clarification for the experiments.**

**Rating:** 6
**Confidence:** 4

**Review:**

The main contribution of this paper is that the authors design a graph deconvolutional network that combines inverse filters in the spectral domain and de-noising layers in the wavelet domain. Further graph autoencoders are proposed based on the graph convolutional networks and the graph deconvolutional networks. Many experiments are done and many previous methods are compared to show the effectiveness of the proposed networks.

My reason for the rating:
Before the experiments part, I think authors clearly present the methods and related works. The results also look quite good compared to other methods. However, I have some questions for the experimental setups that they present at the end of the paper, which somehow confuses me.

Pros:
1. In general, this paper is well-written and the presentation is clear and easy to understand for the method parts.
2. The results well demonstrate the proposed network as reported.

Cons:
I have some comments and concerns for this paper before I made my final decision:
1. It would be great to give a reference to the Maclaurin series approximation for the function of a matrix near (8), or give a simple derivation. Please mention that $n$ starts from 0.
2. For the running time part, it would be great to quantitatively show the time, when choosing a baseline. $10$ and $15$  seem approximation numbers to the reviewer.
3. I recommend the authors explicitly state the previous methods that you have the same setups in the experiments.
4. I find that the authors use $\lambda_{A} = 0$ in the experiments, which means that in (6), the loss function contains only the second term. Could you please clarify this setting?
5. The authors mention that they use best average classification accuracy for all the methods. Could you clarify this part?

---------------------------
After rebuttal

Thank the authors' response to my comments. They also provided new results (i.e., graph generation tasks) to address the Q.4 in my comments. I was about to lower my score when the first reply came as graph generation tasks are just future works in the original paper, but I changed my mind with these results. Although other reviewers may still question the novelty and contributions, I think I would stay with my score according to the comparison and good results reported in the paper. Looking forward to seeing the code someday.

---

> ### Author Response · Authors · 2020-11-19
> **Response to Reviewer4**
>
> We sincerely thank you for the valuable comments. Our responses to the comments are listed below.
>
> Q1. It would be great to give a reference to the Maclaurin series approximation for the function of a matrix near (8), or give a simple derivation. Please mention that  starts from 0.
>
> A1. Thank you for your suggestions. We put a simple derivation in Appendix C as:
> $g_c^{-1}(\lambda_i)$ = $\frac{1}{1-\lambda_i}$ = $\sum_{n=0}^{\infty}\frac{(-1)^{n}n!(-1)^n}{n!}\lambda_i^n$=$\sum_{n=0}^{\infty} \lambda_i^n$,
> where n denotes the n-th order derivative. Thus, Eq.(8) can be obtained as
> $ g_{c}^{-1} * x = U\text{diag}(\sum_{n=0}^{\infty} \lambda_1^n,\ldots,\sum_{n=0}^{\infty}\lambda_N^n)U^\top x=U \sum_{n=0}^{\infty} \Lambda^n U^\top x$.
>
> Q2. For the running time part, it would be great to quantitatively show the time.
>
> A2.  We appreciate the reviewer's suggestions and have revised as follows to address the concerns:  Our model takes 10s to train one epoch of PORTEINS on Tesla P40 24G, while INFOGRAPH needs 127s and MVGRL needs 193s.
>
> Q3. I recommend the authors explicitly state the previous methods that you have the same setups in the experiments.
>
> A3.  If we understand it correctly, “the same setups in the experiments” refers to unsupervised graph-level representation. If so, we do state in the first sentence -- “We adopt the same procedure of previous works (Sun et al., 2020; Hassani \& Khasahmadi,2020) and report the mean 10-fold cross validation accuracy with standard deviation after 5 runs using LIBSVM (Chang & Lin, 2011).”
>
> Q4. $\lambda_A = 0$ in the experiments.
>
> A4.  For the unsupervised graph-level representation and social recommendation, yes, we set $\lambda_A = 0$ and directly use the original graph structure in the decoders of the proposed graph autoencoder framework, i.e., GDN takes the original graph structure as the filter to reconstruct the graph features. In this sense, we don't reconstruct the structure, and the consideration is that both unsupervised graph-level representation experiment and social recommendation can be empowered by reconstructing features. For graph generation tasks, we set $\lambda_A = 1$ and $\lambda_X = 1$.
>
> Q5. The authors mention that they use best average classification accuracy for all the methods. Could you clarify this part?}
>
> A5. In terms of the best average classification accuracy, it means from the combination of hyperparameters such as the dimension of the first layer, dimension of the second layer, the number of clusters K, we report the best results. This follows the settings in Sun et al., (2020) and Hassani \& Khasahmadi,(2020).

---

### Official Review · AnonReviewer3 · 2020-10-29
**The paper proposes a graph deconvolutional network to reconstruct the original graph signal from smoothed node representations obtained by graph convolutional networks.**

**Rating:** 5
**Confidence:** 4

**Review:**

Graph Autoencoders with Deconvolutional Networks

The paper proposes a graph deconvolutional network to reconstruct the original graph signal from smoothed node representations obtained by graph convolutional networks.

The proposed deconvolution incorporates a denoising component based on graph wavelet transforms.

Pros:
-The idea of defining graph deconvolution operators is appealing and and may potentially lead to improvement in the performance of graph reconstruction/generation tasks.
-The Visualization section of the paper shows an advantage of the proposed approach compared to other methods for reconstruction. It maintains information about high-frequency signal.

Cons:
-The potential of a graph deconvolution operator are not fully exploited in the paper, mainly because of the considered tasks that do not require deconvolution because they are not intrinsically reconstruction tasks.
The paper applies the proposed approach to tasks of graph classification in Table 1 and social recommendation (matrix completion) in Table 2. While the comparison with other unsupervised learnimg methods looks favourable to the proposed approach, supervised learning methods are naturally more suited for the tasks in Table 1 and tend to perform slightly better (a comparison with supervised approaches would be appreciated).
Graph Autoencoders are usually applied to tasks of graph generation such as molecule design, where they are one of the most suited approaches. Many works in literature face this problem. A comparison on the generation task would be interesting.
-Considering the Ablation results, the improvement with respect to the ablation approaches seems marginal. Again, my opinion is that the considered tasks are not well suited for the proposed model.
-Hyper-parameter selection: In Appendix A the hyper-parameter selection procedure is not sufficiently detailed. How do you choose the hyper-parameter values? You report the considered ranges but not the procedure you adopt to select them. "parameters of downstream classifiers", in my understanding it refers to the C parameter, or to other hyper-parameters as well?

Minor:
ALATION-GCN -> ABLATION

---Rebuttal--
I acknowledge having checked the authors' response and the revised version of the manuscript, that has been improved since the first revision. Authors did not answer to my request for more details about the hyper parameter selection procedure that has been adopted.

---

> ### Author Response · Authors · 2020-11-19
> **Response to Reviewer3**
>
> We sincerely thank you for the valuable comments. Our responses to the comments are listed below.
>
> Q1. The potential of a graph deconvolution operator are not fully exploited in the paper, mainly because of the considered tasks that do not require deconvolution because they are not intrinsically reconstruction tasks.
>
> A1.  We agree more tasks can be done to fully exploit the potential of GDN and have added graph generation tasks in Section 5.3 in the revision. So far the effectiveness of GDN has been validated in three tasks including unsupervised graph representation, social recommendation and graph generation tasks. Meanwhile, we can't agree graph feature reconstructions (as the cases in unsupervised graph representation and social recommendation) are not reconstruction tasks. In fact, as most previous reconstruction works on graph data focus on topology reconstruction, the potential benefits of feature or signal reconstructions have been largely ignored.
>
> Q2. While the comparison with other unsupervised learnimg methods looks favourable to the proposed approach, supervised learning methods are naturally more suited for the tasks in Table 1 and tend to perform slightly better (a comparison with supervised approaches would be appreciated).
>
> A2.  We add the results of supervised approaches in Table 1. When compared with supervised graph classification models, ours beats the best supervised classification model GIN on IMDB-BIN, on-par with GIN on PROTEINS, IMDB-MULTI, and only loses on REDDIT. We also like to mention that it is unfair for the unsupervised methods to compare with supervised models, as our purpose here is to show that with GDN, graph autoencoders can effectively extract graph representations, and perform better than the other two unsupervised graph representation techniques, i.e., graph kernels and contrastive methods.
>
> Q3. Graph Autoencoders are usually applied to tasks of graph generation such as molecule design, where they are one of the most suited approaches. Many works in literature face this problem. A comparison on the generation task would be interesting.
>
> A3. Great suggestions. As our proposed graph autoencoder framework is not generative model and is not suitable for graph generation task, we validate the proposed GDN in two popular variational autoencoder framework including VGAE (Kipf \& Welling, 2016) and Graphite (Grover et al., 2019). We reconstruct the graph structures using their default methods and the features using GDN. The two reconstructions share the same encoders and sample from the same latent distributions. As shown in Table 3 in the revision, GDN can improve the generative performance of both VGAE and Graphite in general, e.g., it improve the AP score of Graphite by 4.9% on MUTAG , the AUC score of VGAE by 19.4% on PTC-MR and $\log p(A|Z)$ of VGAE by 5.4% on ZINC.
>
> Q4. Considering the Ablation results, the improvement with respect to the ablation approaches seems marginal.
>
> A4. Experimentally, GDN has 1.2% to 1.9% improvements over GCN on all five datasets, which is a very promising result as the competition on these benchmarks is pretty tough. Meanwhile, we would like to mention the results of GDN are the SOTAs of unsupervised graph classification on 4 out 5 datasets. From another perspective, We aim to demonstrate that GDN is better than GCN in reconstructing graph signals by not only retaining low frequency information but also some high frequency information, as gratefully pointed out by you.  Again, as the first work in designing graph deconvolutional networks from inverse filters in spectral domain, our proposed GDN initials a clear way to implement deconvolutional operations and produces a mechanism to prevent the potential noise problem.
>
> Q5. "parameters of downstream classifiers", in my understanding it refers to the C parameter, or to other hyper-parameters as well?
>
> A5.  Yes, "parameters of downstream classifiers" refers to the C parameter. And there is no other hyper-parameters involved.

---

### Official Review · AnonReviewer1 · 2020-10-31
**Incremental work**

**Rating:** 3
**Confidence:** 5

**Review:**

This work considers the graph deconvolution networks. It proposed a graph deconvolutional networks to reconstruct the graph signals. The concerns for this work are as below:

1.	The contributions in this work are incremental. It seems most parts are based on the previous works. The deconvolution part follows the scheme defined in (Bianchi et al. 2020). The overall graph autoencoder also follows a very common settings by involving pooling and unpooling operations. In pooling part, two pooling methods are employed that are hard pooling and soft pooling. However, it is not clear to me how to combine these two methods since they are very different. The equation (3) shows the combination of these two methods but fails to make sense to me. I would like the authors to clarify this and provide detailed explanations for this part.
2.	In both Section 4.1 inverse of GCN and Section 4.2 wavelet de-noising, the authors talked about two methods but didn’t explain why they are related to the proposed graph deconvolution networks. Also, most of the content in these two parts are following previous works like (Li et al. 2019b) and (Xu et al. 2019). I would recommend the authors to clearly point out what is the contribution and how these are related to the proposed methods.
3.	The visualization section in 4.3 is confusion. Firstly, why the authors believe this is a contribution to the graph deconvolutional networks. To my understanding, there is no technical contribution for this paper. It would be better to show this in the experimental parts. Secondly, there is no description on the settings for the illustrations. I didn’t understand how the image are converted into graph and how the inverse GCN and the proposed models are used to them. Thus, the authors need to clarify this and point out the technical contributions for this part.
4.	The experimental results are very limited. The four datasets used in table 1 are very small in terms of the number of graphs, which cannot provide comprehensive evaluations to the proposed results. The baseline methods are not well established. There are far better results on these datasets such as GIN. The authors claim that the unsupervised settings are used. However, it is weird to me why the unsupervised settings do not help the  overall model performances.

---

> ### Author Response · Authors · 2020-11-17
> **Response to Reviewer1(2/2)**
>
> Q3. The visualization section in 4.3 is confusion. Firstly, why the authors believe this is a contribution to the graph deconvolutional networks. To my understanding, there is no technical contribution for this paper. It would be better to show this in the experimental parts. Secondly, there is no description on the settings for the illustrations. I didn’t understand how the image are converted into graph and how the inverse GCN and the proposed models are used to them. Thus, the authors need to clarify this and point out the technical contributions for this part.
>
> A3.  We appreciate this point and agree the settings for the visualization is not very clear. we revise this part and clarify as follow:  (1) the purpose, the experiment targets a traffic network in real service session and is used to provide an intuitive understanding of the difference of GCN, inverse GCN and GDN in reconstructing graph features. It serves as a visual ablation for GDN; and (2) the settings, we select 4438 sensor stations as the node set $V$ and collect their road average occupancy rates. We construct an adjacency matrix A by denoting $A_{ij} = 1$ if sensor station $v_j$ and $v_i$ are adjacent on a freeway. We use the proposed graph autoencoders to reconstruct road occupancy rates in each station, with different variants (GCN, inverse GCN and GDN) working as the decoders. We target traffic network because traffic network can be projected into Euclidean space, which facilitates the public's sense-making. In this vein, we would like to point out there is no conversion from image into graph and the map background is just used for sense-making.
>
> Q4. The experimental results are very limited. The four datasets used in table 1 are very small in terms of the number of graphs, which cannot provide comprehensive evaluations to the proposed results. The baseline methods are not well established. There are far better results on these datasets such as GIN. The authors claim that the unsupervised settings are used. However, it is weird to me why the unsupervised settings do not help the overall model performances.
>
> A4. (1) Comparison with supervised classification models. For the experimental settings of unsupervised graph-level representations, we closely follow the recent methods (Sun et al., 2020; Hassani \& Khasahmadi, 2020) and evaluate the qualify of the unsupervised graph representations in the downstream classification tasks. Thus, it is unfair for these unsupervised methods to compare with supervised models, as our purpose here is to show that with GDN, graph autoencoders can perform better than the other two unsupervised techniques, i.e., graph kernels and contrastive methods.
> Nevertheless, we include the results of popular methods in supervised fashion. The results show ours is better than GIN on IMDB-BIN, on-par with GIN on PROTEINS, IMDB-MULTI, and only loses on REDDIT. (2) The size of datasets. We fully agree the experiments should be done on larger datasets for a fair comparison. That's why we replace two small datasets MUTAG and PTC-MR used in Sun et al., (2020) and Hassani \& Khasahmadi (2020) by PROTEINS and DD, which are larger in terms of both number of graphs and size of graphs. In fact, the largest graph size used in GIN (as raised by the review) is 508 nodes per graph of REDDIT, while the largest graph size used in our experiments is 715 of DD.

---

> ### Author Response · Authors · 2020-11-17
> **Response to Reviewer1(1/2)**
>
> Thanks for your nice review and constructive feedback. Please see our detailed responses below.
>
> Q1. The contributions in this work are incremental. It seems most parts are based on the previous works. The deconvolution part follows the scheme defined in (Bianchi et al. 2020). The overall graph autoencoder also follows a very common settings by involving pooling and unpooling operations. In pooling part, two pooling methods are employed that are hard pooling and soft pooling. However, it is not clear to me how to combine these two methods since they are very different. The equation (3) shows the combination of these two methods but fails to make sense to me. I would like the authors to clarify this and provide detailed explanations for this part.
>
> A1:  Firstly we would like to highlight our main contribution. We study graph deconvolutional networks (GDN) from the perspective of inverse operations in spectral domain (same domain as in Kipf \& Welling, 2017) and identity the inverse operations results in $\textit{high pass}$ and may amplify the noise.  To solve the noise issue,  a de-nosing process based on graph wavelet are further studied. The study on GDN from the perspective of inverse filter, to the best of our knowledge, has not been studied before. Based on GDN, we further make a contribution to graph auto-encoders by reconstructing graph features with deconvolutional layers. Again, we would like to point out that the deconvolutional layer is included in graph auto-encoders for the first time in this work and there is no such component in Bianchi et al. (2020). As for the pooling operations, we don't claim the invention of this operation as we follow Ying et al. (2018). Nevertheless, we provide an explanation as follow: equation (3) strictly follows equation (3)(4) in Ying et al. (2018). Given a soft cluster assignment $S \in \mathbb{R}^{N \times K}$ where $S_{ij}$ represents the probability of node $v_i$ to cluster $j$, $Z = S^{\top}H \in \mathbb{R}^{K \times v}$ derives the feature representations of the coarse graph where each row $Z_j$ represents the feature presentation of cluster $j$, and $A^{pool} = S^{\top}AS$ derives the structure of the coarse graph where each row $A^{pool}_j$ denotes the connectivity of cluster $j$ to all $K$ clusters.
>
> Q2: In both Section 4.1 inverse of GCN and Section 4.2 wavelet de-noising, the authors talked about two methods but didn’t explain why they are related to the proposed graph deconvolution networks. Also, most of the content in these two parts are following previous works like (Li et al. 2019b) and (Xu et al. 2019). I would recommend the authors to clearly point out what is the contribution and how these are related to the proposed methods.
>
> A2: We agree we need to clarify the relation of our components in GDN with previous works and thus have revised the paper accordingly. However, we must point out that our methods have substantial differences with previous works like Li et al. (2019b) and Xu et al. (2019). We address the differences as below: firstly we would like to highlight GDN is a hybrid deconvolutional network which consists of an inverse operator in spectral domain (inverse of GCN in Section 4.1) and de-noising in wavelet domain (wavelet de-noising in Section 4.2). For inverse GCN vs. Li et al. (2019b), the connection of inverse GCN and Li et al. (2019b) is that both follow an analysis framework that categorizes graph neural networks into two stages: a graph convolution stage and a feature transformation stage. Aside from this, inverse GCN and Li et al. (2019b) have distinct differences. Inverse GCN includes a high pass filter and is used to recover graph features from smoothed representations, while Li et al. (2019b) is within the scope of low pass filters and is used to encode smoothed representations. This point is quite obvious as the experiments in Li et al. (2019b) are designed to show it can retain useful representations rather than recovering the original features. For wavelet de-noising vs. Xu et al. (2019), the distinct difference is that we use Maclaurin series approximation to the heat kernel, which has explicit polynomial coefficients and can resemble the heat kernel well when $n =3$ (See Figure 2 in the revision). On the contrary, Xu et al. (2019), though tries to avoid eigen-decomposition by exploiting Chebyshev polynomials, still relies integral operations and there is no clear way that it can scale up.

---

### Author Response · Authors · 2020-11-19
**Summary of revisions made to the paper**

We would like to thank the reviewers for their insightful and helpful comments. We have revised the paper according to the comments. A summary of the major changes is given below:

1. Re-organize the structure of ``Section 4 graph deconvolutional networks. Specifically, we have made the following changes: (1) establish the connection with related works on wavelet neural networks, the distinct difference is that we use Maclaurin series approximation to the heat kernel, which has explicit polynomial coefficients and can resemble the heat kernel well when $n =3$; (2) provide intuitive visualization in Figure 2 for low order Maclaurin Series approximations w.r.t.\ wavelet kernel and inverse wavelet kernel; (3) provide detailed descriptions to the experimental settings of visualization in Section 4.3.  We would like to point out, compared with previous graph convolutional networks that encodes smoothed representations, graph deconvolutional network has substantially difference, and the corresponding solutions are not incremental, as concerned by reviewer1.

2. Add experiments about graph generation. Motivated by the reviewers2' suggestions, we add graph generation tasks in Section 5.3 in the revision. We validate the proposed GDN in two popular variational autoencoder framework including VGAE (Kipf \& Welling, 2016) and Graphite (Grover et al., 2019). We reconstruct the graph structures using their default methods and the features using GDN. The two reconstructions share the same encoders and sample from the same latent distributions. As shown in Table 3 in the revision, GDN can improve the generative performance of both VGAE and Graphite in general, e.g., it improve the AP score of Graphite by 4.9% on MUTAG , the AUC score of VGAE by 19.4% on PTC-MR and $\log p(A|Z)$ of VGAE by 5.4% on ZINC.

3. Provide comparison with supervised graph classification models. We want to point out that comparing unsupervised methods to  with supervised models may not be an apple-to-apple comparison, as our purpose here is mainly to show graph autoencoders performs better in encoding graph representations with GDN in an unsupervised fashion than the other two unsupervised techniques, i.e., graph kernels and contrastive methods. Still, as the reviewer suggested, we include the results of popular supervised methods. The results confirm the proposed GDN outperforms GIN on IMDB-BIN, and is on-par with GIN on PROTEINS, IMDB-MULTI, and only loses on REDDIT.

We submitted a revised version including the aforementioned revisions. Thank you for all the efforts that help us improve the paper.

---

### Decision · Program_Chairs · 2021-01-07
**Final Decision**

**Decision:**

Reject

**Comment:**

The covered topic is timely and of potential impact for many application domains, such as drug design. The paper is well written and presentation is clear. The proposed approach seems to have some degree of originality. Experimental results seem to be generally good, and in the rebuttal the authors have provided further experimental support to their main claim.  There are however some issues that have not been solved by the author’s rebuttal. I think two of them are the most important and related:

i) significance of contribution: although the authors have tried in the rebuttal to explain how the proposed approach differs from related papers, it seems that there are still doubts about the amount of innovation introduced by the paper. This issue could have been mitigated by SOTA experimental results in presence of a proper model selection, that, however does not seem to be the case here (see next point);

ii) model selection: the authors did not clearly explain the model selection procedure in the rebuttal. This is an important issue since it is often easy to get good results by picking the best run a posteriori. Unfortunately in the literature there are highly cited papers where model selection is not performed in a proper way and reviewers very often reject papers just looking at numbers without looking at how the numbers were obtained. So, I believe it is important to accept only papers where model selection is properly done and properly explained, so to allow for reproducibility of experiments.